# An organelle-specific protein landscape identifies novel diseases and molecular mechanisms

Karsten Boldt[1,*], Jeroen van Reeuwijk[2,*], Qianhao Lu[3,4,*], Konstantinos Koutroumpas[5,*], Thanh-Minh T. Nguyen[2], Yves Texier[1,6], Sylvia E.C. van Beersum[2], Nicola Horn[1], Jason R. Willer[7], Dorus A. Mans[2], Gerard Dougherty[8], Ideke J.C. Lamers[2], Karlien L.M. Coene[2], Heleen H. Arts[2], Matthew J. Betts[3,4], Tina Beyer[1], Emine Bolat[2], Christian Johannes Gloeckner[9], Khatera Haidari[10], Lisette Hetterschijt[11], Daniela Iaconis[12], Dagan Jenkins[13], Franziska Klose[1], Barbara Knapp[14], Brooke Latour[2], Stef J.F. Letteboer[2], Carlo L. Marcelis[2], Dragana Mitic[15], Manuela Morleo[12,16], Machteld M. Oud[2], Moniek Riemersma[2], Susan Rix[13], Paulien A. Terhal[17], Grischa Toedt[18], Teunis J.P. van Dam[19], Erik de Vrieze[11], Yasmin Wissinger[1], Ka Man Wu[2], UK10K Rare Diseases Group[#], Gordana Apic[15], Philip L. Beales[13], Oliver E. Blacque[20], Toby J. Gibson[18], Martijn A. Huynen[19], Nicholas Katsanis[7], Hannie Kremer[11], Heymut Omran[8], Erwin van Wijk[11], Uwe Wolfrum[14], François Kepes[5], Erica E. Davis[7], Brunella Franco[12,16], Rachel H. Giles[10], Marius Ueffing[1,*], Robert B. Russell[3,4,*] & Ronald Roepman[2,*]

Cellular organelles provide opportunities to relate biological mechanisms to disease. Here we use affinity proteomics, genetics and cell biology to interrogate cilia: poorly understood organelles, where defects cause genetic diseases. Two hundred and seventeen tagged human ciliary proteins create a final landscape of 1,319 proteins, 4,905 interactions and 52 complexes. Reverse tagging, repetition of purifications and statistical analyses, produce a high-resolution network that reveals organelle-specific interactions and complexes not apparent in larger studies, and links vesicle transport, the cytoskeleton, signalling and ubiquitination to ciliary signalling and proteostasis. We observe sub-complexes in exocyst and intraflagellar transport complexes, which we validate biochemically, and by probing structurally predicted, disruptive, genetic variants from ciliary disease patients. The landscape suggests other genetic diseases could be ciliary including 3M syndrome. We show that 3M genes are involved in ciliogenesis, and that patient fibroblasts lack cilia. Overall, this organelle-specific targeting strategy shows considerable promise for Systems Medicine.

[1] Medical Proteome Center, Institute for Ophthalmic Research, University of Tuebingen, 72074 Tuebingen, Germany. [2] Department of Human Genetics and Radboud Institute for Molecular Life Sciences, Radboud University Medical Center, Geert Grooteplein Zuid 10, 6525 GA Nijmegen, The Netherlands. [3] Biochemie Zentrum Heidelberg (BZH), University of Heidelberg, Im Neuenheimer Feld 328, 69120 Heidelberg, Germany. [4] Cell Networks, Bioquant, Ruprecht-Karl University of Heidelberg, Im Neuenheimer Feld 267, 69120 Heidelberg, Germany. [5] Institute of Systems and Synthetic Biology, Genopole, CNRS, Université d'Evry, 91030 Evry, France. [6] Department of Molecular Epigenetics, Helmholtz Center Munich, Center for Integrated Protein Science, 81377 Munich, Germany. [7] Center for Human Disease Modeling, Duke University, Durham, North Carolina 27701, USA. [8] Department of General Pediatrics, University Children's Hospital Muenster, 48149 Muenster, Germany. [9] German Center for Neurodegenerative Diseases (DZNE) within the Helmholz Association, Otfried-Müller Strasse 23, 72076 Tuebingen, Germany. [10] Department of Nephrology and Hypertension, Regenerative Medicine Center, University Medical Center Utrecht, 3584 CT Utrecht, The Netherlands. [11] Department of Otorhinolaryngology and Donders Institute for Brain, Cognition and Behaviour, Radboud University Medical Center, Geert Grooteplein Zuid 10, 6525 GA Nijmegen, The Netherlands. [12] Telethon Institute of Genetics and Medicine, TIGEM 80078, Italy. [13] Molecular Medicine Unit and Birth Defects Research Centre, UCL Institute of Child Health, London, WC1N 1EH, UK. [14] Cell and Matrix Biology, Inst. of Zoology, Johannes Gutenberg University of Mainz, 55122 Mainz, Germany. [15] Cambridge Cell Networks Ltd, St John's Innovation Centre, Cowley Road, Cambridge, CB4 0WS, UK. [16] Department of Translational Medicine Federico II University, 80131 Naples, Italy. [17] Department of Genetics, University Medical Center Utrecht, 3584 CX Utrecht, The Netherlands. [18] Structural and Computational Biology Unit, European Molecular Biology Laboratory, Meyerhofstrasse 1, 69117 Heidelberg, Germany. [19] Centre for Molecular and Biomolecular Informatics and Radboud Institute for Molecular Life Sciences, Radboud University Medical Center, Geert Grooteplein Zuid 26-28, 6525 GA Nijmegen, The Netherlands. [20] School of Biomolecular & Biomed Science, Conway Institute, University College Dublin, Dublin 4, Ireland. * These authors contributed equally to this work. Correspondence and requests for materials should be addressed to M.U. (email: Marius.Ueffing@uni-tuebingen.de) or to R.B.R. (email: robert.russell@bioquant.uni-heidelberg.de) or to R.R. (email: ronald.roepman@radboudumc.nl).
[#] The members of UK10K Rare Diseases Group have been listed at the end of the paper.

Studies relating genetic variation and biomolecular function[1,2] are often illuminating, but can be hampered by the overall complexity of diseases. Mutations causing the same diseases are often spread across seemingly disconnected cellular processes, meaning that a near-complete understanding of the cell is necessary for a systematic interrogation of disease mechanisms. Such complexity argues that sub-systems, of reduced complexity, could be used as models to develop systematic approaches to study mechanisms of disease. As genome-reduced systems enable Systems Biology[3], isolated systems of reduced complexity, such as organelles where dysfunction leads to one or more diseases, can similarly enable Systems Medicine.

Cilia are spatially and temporally isolated from other cell processes[4] and humans depend on cilia to see, hear, smell, breathe, excrete, reproduce and develop. Mutations disrupting them cause several diseases (ciliopathies) including polycystic kidney disease and other rare disorders like Usher (USH), Bardet-Biedl (BBS), Meckel-Grüber (MKS) and Jeune (JATD) syndromes that are of immense recent biological focus[5]. As many as 1 in 1,000 people are affected by ciliopathies that lead to blindness, deafness, heart failure, diabetes, kidney disease, skeletal defects, infertility and/or cognitive impairment[6]. This has led to a renewed interest in cilia and several efforts to understand these poorly understood organelles.

Studies in animal models and cell culture, show the cilium to be like a cell antenna, harbouring critical components of Shh, Wnt, Hippo, Notch and mTor signalling[7]. Various proteomics and genetics studies have led to lists of proteins likely to reside in the cilia[7–9] though mechanistic details of processes like ciliary transport and proteostasis are unknown, and we still lack a comprehensive picture of the protein machinery operating in cilia.

Here, we employed affinity proteomics to probe the wiring of ciliary proteins and integrated the resulting landscape with disease mutations/variants, cell biology and functional information. The resulting interactome extends knowledge on the ciliary machinery, helps to identify new disease-relevant ciliary proteins and modules, and provides a bounty of new data to aid the understanding, diagnostics and treatment of these devastating genetic disorders.

## Results

**The ciliary landscape.** We determined a ciliary protein landscape by systematic tandem affinity purifications (SF-TAP[10]) coupled to mass spectrometry (MS) for 217 proteins, with known/suspected involvement in ciliary function or disease, in HEK293T cells (Supplementary Fig. 1; Supplementary Data 1), which are ciliated (Supplementary Fig. 2), and an effective means to study cilia[11]. From the selected baits 91 are known ciliopathy genes and 124 are gold-standard ciliary proteins[12]. The 80 baits not in any of these sets are those that frequently appeared in previous ciliary proteomes[8] or were candidate ciliary proteins from previous studies (Supplementary Data 1; Supplementary Data 9). We performed purifications at least twice for 165 baits (644 total) leading to 41,170 bait–prey pairs involving 4,703 proteins (Supplementary Data 2), with reasonable saturation (Supplementary Fig. 3). To identify confident interactions, we adapted the socioaffinity index[13] to account for the partial proteome and weighted protein counts by peptide coverage. Socioaffinity provides a single measure of the association between each pair of proteins based on an entire TAP-data set, considering both the spoke (when one protein retrieves another when tagged) and the matrix (when two proteins are retrieved by another) evidence, and the overall frequency of each protein in the data set. Effectively this gives higher confidence to interactions seen multiple times, and down-weights 'sticky' proteins that are often

seen. Benchmarking these values with known interactions and a set of negative interactions[14] gave excellent sensitivity and provided false-positive and false-discovery rates (FPR, FDR) that gave confidence intervals (Supplementary Fig. 4; Supplementary Data 3). We identified complexes of 3–20 subunits by clustering the interactions using clique identification (Supplementary Data 4).

The landscape includes 1,319 proteins and 4,905 interactions (FDR/FPR ≤ 0.1), including 91 of 154 known ciliopathy genes, 134 of 302 gold-standard ciliary proteins and 84 of 362 recently identified ciliary proteins[9]. Our approach shows power in identifying real ciliary components as 16 ciliopathy genes, 23 gold-standard and 53 ciliary proteins not among our original baits were nonetheless found (Supplementary Data 9). The socioaffinity index has, as expected[13], removed interactions likely to be the result of missed contaminants or very high protein abundances. Specifically, the 16 proteins with the highest (top 0.1%) median human protein abundancies[15] are found multiple times across a total of 619 (96.2% of the total) purifications, but only one (vimentin) has any significant interactions in our network, and the best two (of 12) of these are known interactions with NEFM/NEFL.

Clustering of these interactions yielded 52 complexes involving 359 proteins distributed across ciliary and other cellular processes (Fig. 1; Supplementary Fig. 1; Supplementary Data 4). Twenty-four complexes have significant overlaps with known complexes (whether ciliary or not), of which 16 contain canonical gold-standard[12] ciliary components. The remaining 28 are largely novel, of which 15 contain one or more known ciliary proteins. Known complexes include those in ciliary transport (IFT-A and -B, the BBSome and KIF3 complex), organellar organization/ transport (the exocyst, dynactin and dynein), centriole/basal-body organization (MKS1) and several other not previously associated with ciliary function (below). Interconnections between ciliary transport and cytoskeleton/centrosome complexes, supports the view that canonical ciliary proteins have roles outside the cilium[16]. We defined core and attachment[13] subunits for most complexes (Supplementary Data 4). For instance, the GTPase RALB and BLOC1S2 are known attachments of the exocyst and IFT-B[17,18] complexes, respectively (Fig. 2a,d). Finer structure for complexes is also apparent (Fig. 2), including known exocyst sub-complexes[17], the known sub-network involving the progression of NPHP and RPGR proteins (Fig. 2f)[19], and new sub-complexes in IFT-B (below).

Although we did not determine stoichiometries of the complexes, comparison of known protein levels across many cell types[15] shows that they are nevertheless stoichiometrically logical: there are few complex cores where one component has a wildly different abundance from the others. Overall, the median differences in abundancies are significantly lower (t-test $P < 0.0001$) when looking at proteins within complexes (23.6 p.p.m.) compared with those between complexes (126 p.p.m.) or involving proteins that were detected in the screen but not in any significant interactions (140 p.p.m.). This suggests that the socioaffinity filtering is effective at removing non-specific components and identifying complexes that are stoichiometrically sensible.

**Proteins and complexes essential to ciliary function.** Among the newly identified complexes, several involve multiple proteins not previously described to act together. For example, we see the ciliogenesis transcription factor FOXJ1 (Fig. 2g) in a complex with Polo-like kinase 1 and the cilia- and flagella-associated protein 20 (CFAP20). This complex interacts with another

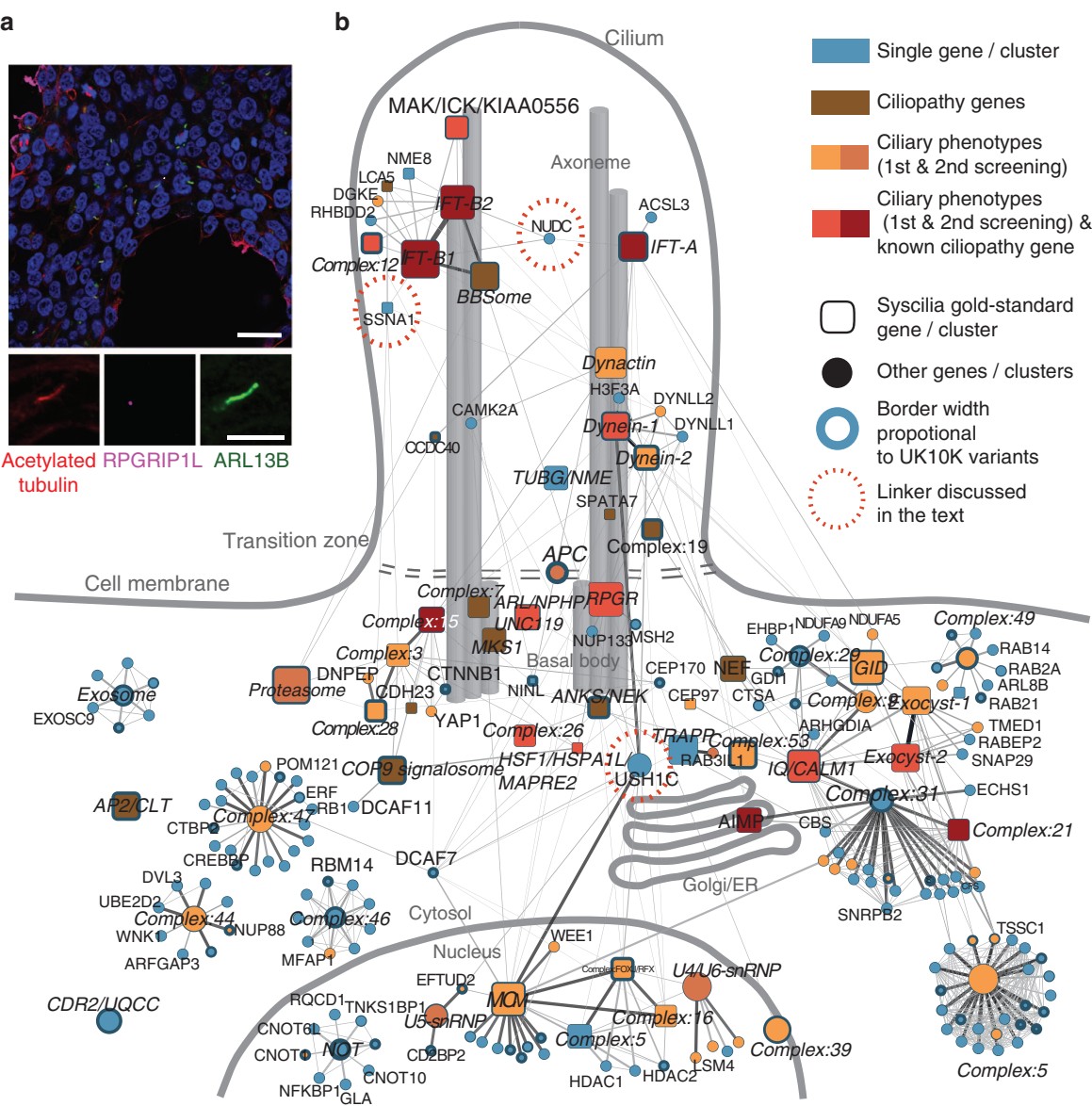

**Figure 1 | Overview of the ciliary landscape. (a)** HEK293T cells stained with the ciliary marker ARL13B (green), the transition zone marker RPGRIP1L (purple), and the axonemal marker acetylated alpha-tubulin (red). Scale bar, 20 μm. In the magnifications the scale bar represents 5 μm. **(b)** Complexes/proteins identified in this study are depicted by circles and rounded boxes. Rounded boxes show complexes/proteins in the Syscilia gold-standard ciliary proteins. The edge thickness is proportional to the socioaffinity index, and proteins/complexes are coloured according to whether they have ciliary phenotypes. The border thickness is proportional to the number of variants in UK10K ciliopathy patients.

containing the X-box factors, RFX1-3, and HDAC1 and 2. Other proteins co-purified with this complex, for example, forkhead box proteins, TBC1D32/broad minded and CDK20/cell cycle-related kinase, suggesting they act directly on the transcriptional regulation of ciliogenesis, which explains their proposed role in coordination of ciliary assembly[20]. We also see the ciliary protein KIAA0556, recently associated with Joubert syndrome, in complex with kinases ICK and MAK, the latter of which interacts with IFT-B (Fig. 1b), supporting a role in the IFT-B train[21].

Seventeen ciliary proteins (including six IFT subunits) retrieved subunits of the glucose-induced deficiency (GID) RING E3 ubiquitin ligase complex, involved in regulating gluco-neogenesis[22], and tagged GID subunits retrieved 18 ciliary proteins, suggesting a ciliary role for this complex. We found that GID complex components localized to the ciliary base in both brain and kidney tissue (Fig. 2d) suggesting a role in cellular energy homoeostasis in cilia. A general role for the

ubiquitin–proteome system in cilia[23] is also supported by the presence of the anaphase-promoting complex, the proteasome and eight ubiquitin conjugating/modifying enzymes in our network (Fig. 2e); absence of several of these proteins also disrupts ciliogenesis[24].

Overall 1,008 of the 1,319 proteins found in our landscape are not known to be ciliary, though we expect several of these to play non-ciliary roles. More stringently, 544 non-ciliary proteins are either in a complex and/or a confident interaction (FDR/FPR ≤ 0.1) with gold-standard proteins (Supplementary Data 5) of which 77 have an siRNA-induced ciliary phenotype[24] and 32 are among 331 novel (out of 371 total) ciliary localized proteins from a recent proteomics study[9]. For 39 there is at least one homozygous missense variant in the UK10K ciliopathies[25] data set (377 have heterozygous variants). This subset is an excellent starting point for new investigations into ciliary function and disease.

**Architecture of intraflagellar transport complexes.** Despite an established functional connection related to ciliary transport, we saw no significant direct physical connections between IFT-A and B. This is in broad agreement with what is currently known as, despite some early, and partly indirect, evidence of a physical association[26], they are not normally seen to interact[27]. There are also comparatively few other proteins that bridge the IFT-A and B complexes. Apart from the known linker LCA5 (ref. 11), the only connection we found between them is NUDC, a WD-repeat, beta-propeller-specific co-chaperone[28]. Within our un-processed

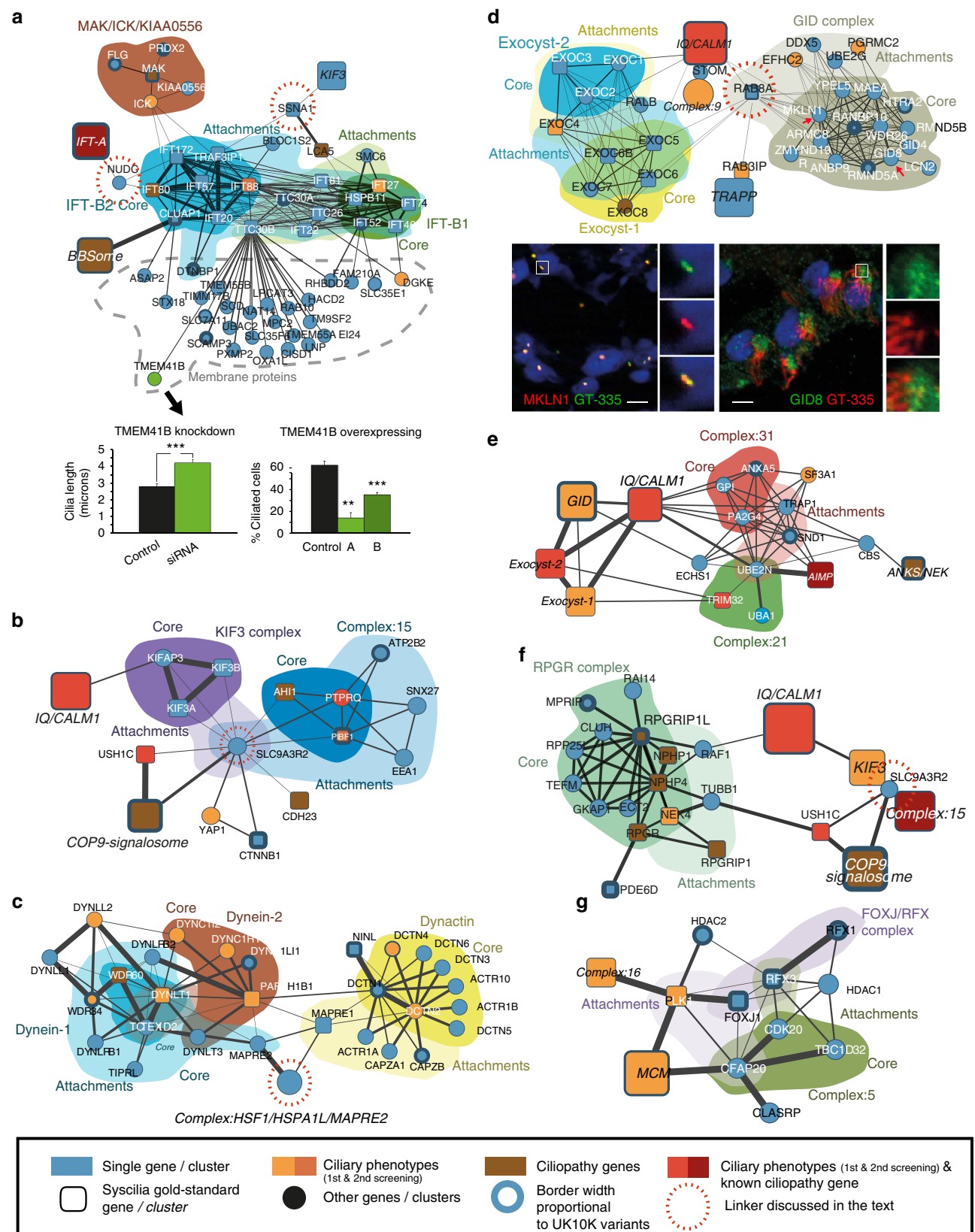

purification data, 13/25 proteins that retrieve NUDC (when tagged and overexpressed) contain WD-repeats (Fisher test $P < 0.0001$; Supplementary Fig. 5), but only 2/37 proteins retrieved by tagged NUDC contain WD-repeats, which is roughly what one would expect from a chaperonin. Interestingly, when we searched for Gene ontology terms enriched among 52 proteins targeted by NUDC[28], only ciliary, axoneme, centrosome, and ubiquitination processes were significant, suggesting this co-chaperone to be particularly functionally relevant for cilia. The presence of this co-chaperone is not indicative of non-specific chaperone proteins (or other parts of the protein synthesis or maintenance machinery) in our network, as these are effectively filtered by the socioaffinity metric as observed previously[13]. For example, CCT has been proposed to be involved in BBSome assembly[29]. We see CCT subunits in 449 purifications, though their promiscuity means that all 16,331 possible interactions are insignificant with just nine marginally significant (FPR/FDR $\leq 0.2$) interactions all involving known ciliary proteins, including BBS5 and BBS4.

The IFT-B particle appears to consist of two sub-complexes (Fig. 3), with IFT88 at the interface. These correspond to core (IFT-B1), and peripheral subunits described previously[30], though with the latter forming a distinct complex (IFT-B2). Sucrose density centrifugation and EPASIS[31] analyses support this finding (Fig. 3; Supplementary Fig. 6; Supplementary Data 6). Additionally, we used structural and interaction information[32,33] to identify rare IFT-B missense variants (identified by targeted resequencing of severe ciliopathy cases), that might affect interactions in IFT-B (Supplementary Fig. 7) and potentially contribute to disease severity. Six out of 10 predicted interaction targeting variants could be purified from HEK293T cells and compared to wild type (Supplementary Fig. 1; Supplementary Data 7). Three of them specifically affected one sub-complex (Fig. 2a; Supplementary Fig. 8). For example, IFT88 p.R607H, a heterozygous variant in an MKS fetus, leads to a specific loss of IFT-B1, supporting IFT88 as a bridge between IFT-B1/B2, and suggesting that this residue might mediate interactions with IFT-B1. There is no evidence that these variants are recessive or disease-causing alleles. We expect that they are modifiers affecting disease severity (for example, as a result of mutational load)[34], and further tests can establish the impact in the context of causal loci. Regardless of their ultimate genetic meaning, these observations provide additional support that IFT-B forms two sub-complexes.

IFT-B components IFT20 and TRAF3IP1 interact with Dysbindin-1 and BLOC1S2 (Fig. 2a), components of the BLOC-1 complex, involved in the transport of membrane cargos and endosomal trafficking. The association of IFT-B components with periciliary cytoplasm membrane vesicles in dendrites supports this link[35], and the IFT involvement in vesicle transport is

corroborated by its likely protocoatomer origin[36]. Possibly related to this, TTC30B (IFT-B) interacts with 20 functionally diverse transmembrane proteins (Fig. 2a), including the uncharacterized TMEM41B, which shows a ciliary phenotype: siRNA downregulation increases and overexpression decreases ciliary length (Fig. 2a; Supplementary Fig. 9). The lack of functional

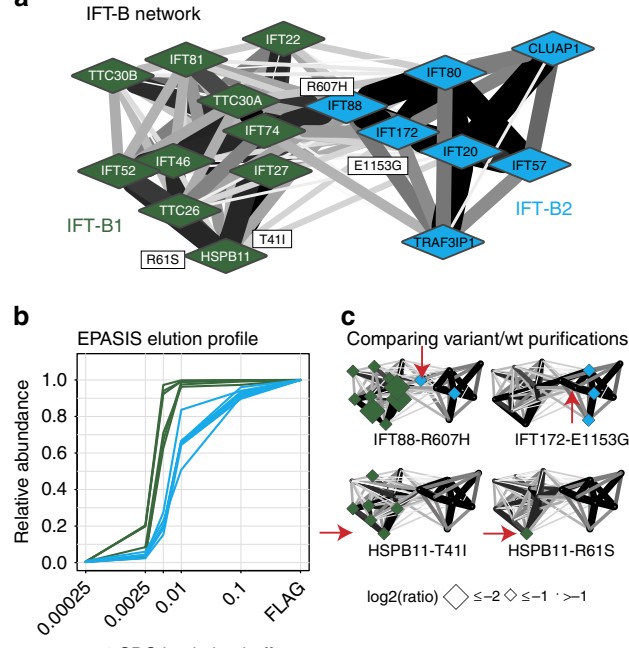

**Figure 3 | Identification of IFT-B sub-complexes and edgetic variants.** (**a**) Socioaffinity-weighted, spring-embedded (cytoscape) layout of IFT-B proteins with two sub-complexes indicated. (**b**) Cumulative elution profiles for IFT-B1/B2 proteins FLAG-purified and analysed by EPASIS in HEK293 cells stably expressing IFT88 or IFT27. Green and blue lines show components of IFT-B1 and -B2 sub-complexes, respectively. (**c**) Networks showing protein depletions in IFT-B comparing mutant to wild type with TAP-MS. Red arrows denote proteins with variants, and protein size is proportional negative fold-change. Top left, IFT88 p.R607H, a heterozygous MKS patient variant leads to a loss of IFT-B1. Bottom left, HSPB11 p.T41I (heterozygous MKS) at the IFT27 interface, leads to the loss of IFT-B1. Top right, IFT-B2 subunit, IFT172 p.E1153G (heterozygous in MKS) leads to a loss of IFT-B2. Bottom right, HSPB11 p.R61S (heterozygous JATD), on the surface, potentially interacting with an unknown partner, though not at any known interface affects only HSPB11 itself. Green and blue nodes represent components of the IFT-B1 and -B2 sub-complexes, respectively.

**Figure 2 | Complexes and networks within the landscape.** (**a**) Detailed network of IFT-B1/2 and MAK/ICK/KIAA0556; IFT-B is linked to IFT-A by NUDC, and to complex KIF3 by SSNA1. The IFT-B protein TTC30B interacts with multiple membrane proteins. One of those, TMEM41B, was further analysed and shows a ciliary length phenotype upon modulation of expression by siRNA knock-down and overexpression. For both, knockdown and overexpression, biological triplicates were analysed and a t-test was performed. P values below 0.01 are represented by ** and below 0.001 by ***. Error bars represent the s.e.m. (**b**) Detailed interaction network of the KIF3 complex and Complex:15, with SLC9A2R2 bridging ciliary processes. (**c**) Detailed interactions between Dynein and Dynactin intermediated by the HSF1/HSPA1L/MAPRE2 linker complex. (**d**) Muskelin/RanBP9/CTLH complex (GID complex in Yeast) network showing core, attachments and links to several other complexes, mediated by RAB8A. Immunofluorescence demonstrates the localization of two GID components, GID8 and MKLN1 (red arrows in the network) to the ciliary base: MKLN1 in kidney tubule epithelial cells (anti-MKLN1, left panel, red); GID8 (right panel, green) in multi-ciliated brain ependymal cells. DAPI staining (blue) marks the nucleus, GT335 co-staining (green or red) marks the cilium. Scale bars represent 10 μm. (**e**) Complex:21 and 31 containing several ubiquitin conjugating or modifying enzymes in interaction with the GID and exocyst complexes. (**f**) Elaborated view of the sub-network involving the NPHP1-NPHP4-RPGRIPL/PDE6D/RPGR complex, and its association with the complexes IQ/CALM1, KIF3, COP9 signalosome, and Complex:15. (**g**) Ciliogenesis transcription factor FOXJ1 stably interacts with PLK1 (Polo-like kinase 1) and CFAP20, and is linked to the FOXJ/RFX complex and Complex:15.

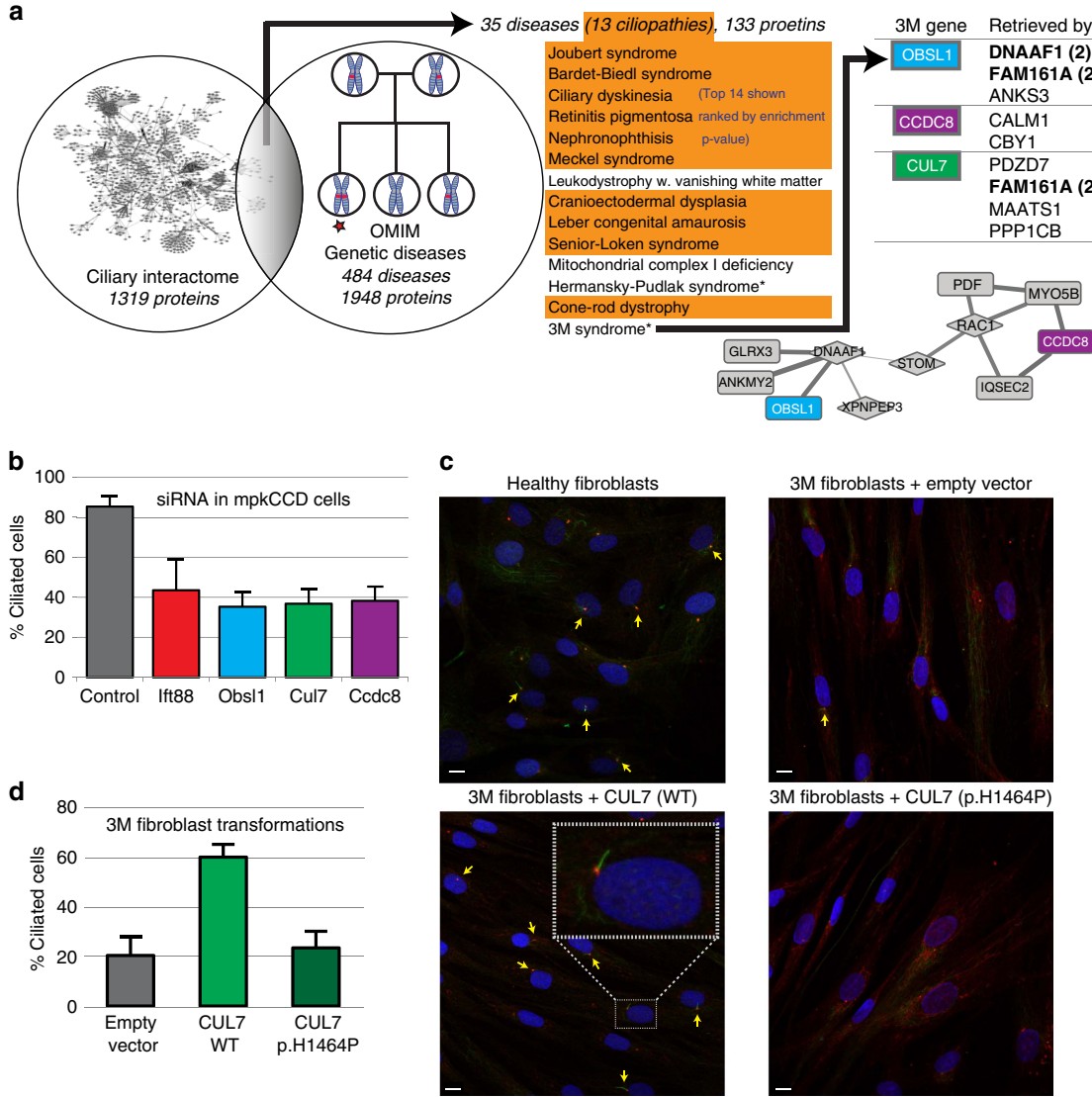

**Figure 4 | 3M Syndrome is a ciliopathy. (a)** Schematic showing comparison of interactome to disease genes with the top scoring diseases shown (orange denotes known ciliopathies). The three 3M associated proteins are shown right, with the baits that retrieved them, and within the network (below). Matrix relationships are not shown in the table. **(b)** siRNA down-regulation of these genes in mpkCCD cells reveals that 3M genes are involved in ciliogenesis (knockdown of *IFT88*, known to affect ciliogenesis, is shown for comparison). **(c)** Fibroblasts from a 3M patient have fewer cilia than controls. The transition zone marker MKS1 is shown in red, acetylated tubulin in green and nuclear staining with DAPI in blue. Scale bars represent 5 μm. **(d)** Quantification of differences in ciliated cell count comparing 3M fibroblasts to those transformed with wild-type or mutant (p.H1464P) CUL7 or empty vector. Wild-type CUL7 restored ciliogenesis, while mutant CUL7 did not. **(b–d)** For all experiments biological triplicates with technical duplicates were performed. Error bars represent the s.e.m.

commonalities among these membrane proteins raises the possibility that they could be IFT-B cargos targeted by TTC30B.

**Proteins bridging ciliary complexes.** Several proteins appear to link important ciliary complexes or proteins (Fig. 1). For example, SSNA1, potentially in collaboration with LCA5, both have ciliary transport phenotypes[11] and link IFT-B to Dynein and KIF3 complexes. We also see RAB8A interacting with RAB3IP and the TRAPP complex as known[37], but also with the GID and exocyst complexes and the membrane protein stomatin, involved in the formation of membrane protrusions (Fig. 2d), suggesting new roles in membrane protein trafficking[38,39]. SLC9A3R2, a scaffold protein not associated with ciliary function but which interacts with seven gold-standard proteins (Fig. 2b), interacts with known partners YAP1 and CTNNB1 plus several proteins involved in Usher syndrome and non-syndromic deafness and the COP9

signalosome (Fig. 2b) that hint at roles in actin attachment/polarization[40], DNA damage response or proteasomal degradation[41]. Finally, there are several proteins linking complexes to the kinetochore, such as microtubule-associated protein RP/EB family members 1 and 2 (MAPRE1,2) and platelet-activating factor acetylhydrolase IB subunit alpha (PAFAH1B1), which lie between IFT-B, GID and the dynactin and dynein kineotochore/microtubule complexes (Fig. 2c).

**Comparison with previous studies.** The BioPlex data set[42], which currently contains 5,087 affinity purifications from HEK293T cells, has 81 of the 217 baits we tagged here. Calculating socioaffinities gives 63,018 confident interactions in BioPlex, of which 421 overlap with our 4,905 interactions (considering the entire BioPlex bait–prey pairs the overlap is 271). Another recent study in HeLa cells involving 1,125

GFP-tagged proteins coupled to quantitative proteomics identified 31,944 significant protein interactions[43] also with a low overlap: 239 common interactions, principally involving the GID, dynactin, CNOT, MCM and exosome complexes (which contain the 28 common baits). The low overlaps highlight the need for specific sub-proteome targeting to uncover interactions of interest as targeting a small subset of baits is insufficient to resolve complexes and interactions fully. The subtleties of architecture that we see in the IFT-B and exocyst complexes are also not apparent in the BioPlex set (and disappear when we simulate fewer baits/repetitions; Supplementary Figs 10 and 11), highlighting the value of repeated reverse tagging to provide high-resolution interactomes.

**New ciliopathies emerging from the organellar landscape**. The variants affecting IFT-B (above) illustrate how genetic changes can inform mechanistic biology. A natural question is whether this works in reverse: can our ciliary landscape inform clinical genetics? Our interactome (1,319 proteins) overlaps with gene sets from 35 genetic diseases (Fig. 4) of which 11 are known ciliopathies (Supplementary Data 8), and others are ciliopathy related (for example, 'Deafness' or 'Mental Retardation') and several, including Amyotrophic lateral sclerosis, Hermansky-Pudlak, Nephrotic and 3M syndromes are potentially new ciliopathies. We did not tag any of the three known 3M syndrome ('3M complex'[44]) -associated proteins though two (OBSL1 and CCDC8) are in our landscape, and the third (CUL7) was detected in multiple purifications, but made no confident interactions. siRNA knock-downs (validated by qPCR; Supplementary Fig. 12) reduced ciliated fractions in mpkCCD cells, which could be rescued by co-expressing human orthologs (Fig. 4; Supplementary Fig. 12). Fibroblasts from a skin biopsy of a 3M case (CUL7 homozygous mutations), known to disrupt ubiquitination[45], had significantly fewer cilia than controls (Fig. 4; Supplementary Fig. 13; ciliary length unchanged). Cilia could also be restored by overexpression of wild type, but not mutant CUL7 (Fig. 4). This ciliary phenotype suggests that 3M syndrome is indeed a ciliopathy.

**Discussion**

The rich landscape including new ciliary-associated proteins, interactions and complexes, when coupled to the growing array of genetic and functional information, will undoubtedly lead to many additional insights into ciliary function and disease. Our organelle-specific interactome also shows considerable power to suggest new genetic diseases (for example, 3M syndrome) likely related to ciliary dysfunction. Identifying novel ciliopathies can have an immediate impact on diagnostics and treatments. For instance, diagnoses can be aided by examining ciliary frequency in young patients' cells[46].

Targeted, repeated, reverse-tagged, TAP-MS proteomics coupled to socioaffinity uncovers physically meaningful interactions not always apparent in high-throughput studies[42,43]. Moreover, the success of this strategy at uncovering finer sub-structures has certain implications for structural biology. Indeed, since the acceptance of this manuscript a discrete structure for many of our IFT-B2 components has provided additional support for this approach[47]. Whole proteome data of this quality could provide unprecedented insights into the architecture of many additional protein complexes. The *edgetic*[48] disease variants affecting specific sub-complexes also shows the complementarity of genetic and mechanistic investigations. A larger study of >1,000 disease mutations has shown that many affect protein–protein interactions[1] and additional studies like that we have performed here, in concert, for example, with larger complex data

sets[42], could also illuminate more generally how disease variants impact protein function.

Overall, this study has demonstrated the great complementarity of proteomics and genetics and the power of focussing on a disease-relevant organelle. As such, this work provides a framework for powerful future applications in biomedicine.

## Methods

**Affinity purifications.** To determine the ciliary protein network, we selected a set of 217 proteins, among which 124 are Syscilia gold standard proteins, 91 are ciliopathy-associated proteins, and 80 are proteins with predicted ciliary function. The proteins were overexpressed in HEK293T cells, fused to a SF-TAP tag to enable tandem affinity purification of the associated protein complexes. The cells were lysed and after clearance of the lysate by centrifugation, the lysate was subjected to a two-step purification via the StrepII-tag, followed by an enrichment using the FLAG moiety. Competitive elution was achieved by addition of the FLAG peptide. The eluate was precipitated by methanol–chloroform and then subjected to mass spectrometric analysis.

**Mass spectrometry.** Following precipitation, SF-TAP-purified complexes were solubilized and proteolytically cleaved using trypsin. The resulting peptide samples were desalted and purified using stage tips before separation on a Dionex RSLC system. Eluting peptides were directly ionized by nano-spray ionization and detected by a LTQ Orbitrap Velos mass spectrometer. Mascot was used to search the raw spectra against the human SwissProt database for identification of proteins. The Mascot results were post validated by Scaffold which employs the protein prophet algorithm.

Identification and label-free quantification for EPASIS and sucrose density centrifugation data was performed with MaxQuant. The peptide and protein false-discovery rates were set to 1% and only unique peptides were used for quantification.

**Network and complex delineation.** We modified the socioaffinity metric[13] to consider protein coverage and to account for the lack of complete proteome tagging. We computed false-positive and false-negative rates using a gold standard of known interactions and a systematically derived set of negative interactions. We applied a Hierarchical Clique Identification approach to cluster proteins and defined attachments as proteins having at least two significant links to the cluster (without being in the cluster itself).

**Sub-complex analysis.** For both, sucrose density gradient centrifugation and EPASIS, the SF-TAP-tagged bait proteins were stably expressed in HEK293 cells and the complexes were affinity purified by FLAG purification. For sucrose density centrifugation, the complexes were eluted by addition of FLAG peptide, and sub-complexes were separated by a discontinuous gradient and fractionated after centrifugation at $166,000g_{AV}$ (ref. 49). The fractions were precipitated and subjected to label-free mass spectrometric analysis.

EPASIS makes use of controlled destabilization of protein–protein interactions by the addition of low concentrations of SDS (Supplementary Fig. 6). The purified complexes were immobilized on FLAG beads. By applying a step gradient, interactions of bait protein and sub-complexes are sequentially destabilized and thereby sub-complexes eluted. Each fraction was subjected to label-free quantification by mass spectrometry before the quantitative data were used to calculate elution profile distances to detect co-eluting sub-complexes.

**Affinity purification.** In total 217 Strep-FLAG tandem affinity purification (SF-TAP)[10] expression constructs were generated (Supplementary Data 1). Bait protein selection was based on the association of proteins with ciliopathies (including mutant vertebrates showing ciliopathy features) or involvement in IFT. In addition, we selected part of our candidate list of ciliary proteins which is a compilation of a subset from the ciliary proteome database[8] (type: non-reciprocal; e-value cut-off: 1E − 10; study selection all, ≥4 studies) and candidate ciliary proteins resulting from previous studies in our labs. Gateway-adapted cDNA constructs were obtained from the Ultimate ORF clone collection (Thermo Fisher Scientific) or generated by PCR from IMAGE clones (Source BioScience) or human marathon-ready cDNA (Clontech) as template and cloning using the Gateway cloning system (Thermo Fisher Scientific) according to the manufacturer's procedures followed by sequence verification.

HEK293T cells were grown in DMEM (PAA) supplemented with 10% fetal bovine serum and 0.5% penicillin/streptomycin. Cells were seeded, grown overnight and then transfected with the corresponding SF-TAP-tagged DNA constructs using PEI reagent (Polysciences) according to the manufacturer's instructions. Forty-eight hours later, cells were harvested in lysis buffer containing 0.5% Nonidet-P40 (NP-40), protease inhibitor cocktail (Roche), and phosphatase inhibitor cocktails II and III (Sigma-Aldrich) in TBS (30 mM Tris-HCl, pH 7.4 and 150 mM NaCl) for 20 min at 4 °C. Cell debris and nuclei were removed by centrifugation at 10,000g for 10 min.

For SF-TAP analysis, the cleared supernatant was incubated for 1 h at 4 °C with Strep-Tactin superflow (IBA). Subsequently, the resin was washed three times in wash buffer (TBS containing 0.1% NP-40 and phosphatase inhibitor cocktails II and III, Sigma-Aldrich). Protein baits were eluted with Strep-elution buffer (2 mM desthiobiotin in TBS). For the second purification step, the eluates were transferred to anti-Flag M2 agarose (Sigma-Aldrich) and incubated for 1 h at 4 °C. The beads were washed three times with wash buffer and proteins were eluted with FLAG peptide (200 µg ml$^{-1}$, Sigma-Aldrich) in TBS. After purification, the samples were precipitated with chloroform and methanol and subjected to in-solution tryptic cleavage[50]. Precipitated protein samples were dissolved in 30 µl, 50 mM ammonium bicarbonate (Sigma-Aldrich), supplemented with 2% RapiGest (Waters) before 1 µl 100 mM DTT (Merck) was added. After incubation at 60 °C for 10 min, 1 µl, 300 mM 2-iodacetamide was added followed by incubation at room temperature for 30 min in the dark. Before overnight incubation at 37 °C, 1 µg of trypsin (Sigma-Aldrich, sequencing grade) was added. The reaction was stopped by addition of trifluoracetic acid to a final concentration of 1%.

**Generation of stable cell lines.** For stable HEK293 cells, cells were cultivated as indicated above and transfected with the corresponding DNA construct using PEI reagent. After 48 h, the medium was exchanged by growing medium supplemented with G418 (Biochrom, 750 mg ml$^{-1}$). The cells were cultivated for ~3 weeks, until the transiently transfected cells died. The medium was exchanged regularly to ensure normal growth. Afterwards, the cells were split in a ratio of 1:100 and cultivated until single colonies were observed. Colonies were transferred to six-well plates and cultivated to confluency. For evaluation, a part of the cells was lysed and applied to western blot analysis, using an anti-FLAG-M2-HRP antibody (Sigma-Aldrich) for detection of the expressed SF-TAP-fusion protein.

**Mass spectrometric analysis.** *Qualitative mass spectrometry.* After precipitation of the proteins by methanol–chloroform, a tryptic in-solution digestion was performed as described above[50]. LC–MS/MS analysis was performed on a NanoRSLC3000 HPLC system (Dionex) coupled to a LTQ or to a LTQ Orbitrap Velos mass spectrometer (Thermo Fisher Scientific) by a nano-spray ion source. Tryptic peptide mixtures were automatically injected and loaded at a flow rate of 6 µl min$^{-1}$ in 98% buffer C (0.1% trifluoroacetic acid in HPLC-grade water) and 2% buffer B (80% acetonitrile and 0.08% formic acid in HPLC-grade water) onto a nanotrap column (75 µm i.d. × 2 cm, packed with Acclaim PepMap100 C18, 3 µm, 100 Å; Dionex). After 5 min, peptides were eluted and separated on the analytical column (75 µm i.d. × 25 cm, Acclaim PepMap RSLC C18, 2 µm, 100 Å; Dionex) by a linear gradient from 2 to 35% of buffer B in buffer A (2% acetonitrile and 0.1% formic acid in HPLC-grade water) at a flow rate of 300 nl min$^{-1}$ over 33 min for EPASIS samples, and over 80 min for SF-TAP samples. Remaining peptides were eluted by a short gradient from 35 to 95% buffer B in 5 min. The eluted peptides were analysed by using LTQ Orbitrap XL, or a LTQ OrbitrapVelos mass spectrometer. From the high-resolution mass spectrometry pre-scan with a mass range of 300–1,500, the 10 most intense peptide ions were selected for fragment analysis in the linear ion trap if they exceeded an intensity of at least 200 counts and if they were at least doubly charged. The normalized collision energy for collision-induced dissociation was set to a value of 35, and the resulting fragments were detected with normal resolution in the linear ion trap. The lock mass option was activated and set to a background signal with a mass of 445.12002 (ref. 51). Every ion selected for fragmentation was excluded for 20 s by dynamic exclusion.

For qualitative results the raw data were analysed using Mascot (Matrix Science, version 2.4.0) and Scaffold (version 4.0.3, Proteome Software). Tandem mass spectra were extracted, charge state deconvoluted and deisotoped by extract_msn.exe version 5.0. All MS/MS samples were analysed using Mascot. Mascot was set up to search the SwissProt_2012_05 database (selected for Homo sapiens, 2012_05, 20,245 entries) assuming the digestion enzyme trypsin. Mascot was searched with a fragment ion mass tolerance of 1.00 Da and a parent ion tolerance of 10.0 p.p.m. Carbamidomethyl of cysteine was specified in Mascot as a fixed modification. Deamidation of asparagine and glutamine and oxidation of methionine were specified in Mascot as variable modifications. Scaffold was used to validate MS/MS based peptide and protein identifications. Peptide identifications were accepted if they could be established at >80% probability by the Peptide Prophet algorithm[52] with Scaffold delta-mass correction. Protein identifications were accepted if they could be established at greater than 95.0% probability and contained at least two identified peptides. Protein probabilities were assigned by the Protein Prophet algorithm[53]. Proteins that contained similar peptides and could not be differentiated based on MS/MS analysis alone were grouped to satisfy the principles of parsimony. Furthermore, proteins were only considered to be specific protein complex components if they were not detected in the control experiments.

Data were exported from Scaffold (Proteome Software) to tab-delimited protein reports and curated into data templates for database integration with other data and further analysis. Although great care was taken to avoid sample carryover during the experimental procedure of TAP and MS analysis, we noted occasionally carryover of bait proteins in a series of TAP experiments analysed consecutively by MS. Therefore we removed all bait proteins per series of experiments from the MS results (127 protein identifications in total) and experiments were replicated for known IFT and ciliopathy-associated proteins making sure that bait proteins were

in unique combinations in new series of experiments. This allowed us to detect protein interactions between proteins that are both bait and prey proteins in our experiments.

*Quantitative mass spectrometry.* For quantitative analysis, MS raw data were processed using the MaxQuant software (version 1.5.0.3 (ref. 54)). Trypsin/P was set as cleaving enzyme. Cysteine carbamidomethylation was selected as fixed modification and both methionine oxidation and protein acetylation were allowed as variable modifications. Two missed cleavages per peptide were allowed. The peptide and protein false-discovery rates were set to 1%. The initial mass tolerance for precursor ions was set to 6 p.p.m. and the first search option was enabled with 10 p.p.m. precursor mass tolerance. The fragment ion mass tolerance was set to 0.5 Da. The human subset of the human proteome reference set provided by SwissProt (Release 2012_01 534,242 entries) was used for peptide and protein identification. Contaminants like keratins were automatically detected by enabling the MaxQuant contaminant database search. A minimum number of 2 unique peptides with a minimum length of seven amino acids needed to be detected to perform protein quantification. Only unique peptides were selected for quantification. For label-free quantification the minimum LFQ count was set to 3, the re-quantify option was chosen. The option match between runs was enabled with a time window of 2 min, fast LFQ was disabled.

**Network and complex delineation.** *Socioaffinity index and definition of thresholds.* The TAP-MS data includes baits preys together with the unique peptide counts and the sequence coverage for each protein identified. Before any consideration we removed a set of potential/known contaminant proteins (ALB, CALD1, CDSN, DCD, DSP, DSC1, DSC2, DSC3, DSG1, DSG2, DSG3, DSG4, EOMES, EPPK1, EVPL, EVPL, GSDMA, GSDMB, GSDMC, GSDMD, HRNR, KRT10, KRT12, KRT13, KRT14, KRT15, KRT16, KRT17, KRT18, KRT19, KRT20, KRT23, KRT24, KRT25, KRT26, KRT27, KRT28, KRT39, KRT40, KRT9, KRT31, KRT32, KRT2, KRT76, KRT77, KRT1, KRT3, KRT4, KRT5, KRT6A, KRT6B, KRT6C, KRT71, KRT72, KRT73, KRT74, KRT75, KRT78, KRT79, KRT7, KRT80, KRT8, KRTCAP3, KCT2, KPRP, KRTAP10-1, KRTAP10-2, KRTAP10-3, KRTCAP3, KCT2, KPRP, KRTAP10-1, KRTAP10-2, KRTAP10-3, KRTAP10-4, KRTAP10-5, KRTAP10-6, KRTAP10-7, KRTAP10-8, KRTAP10-9, KRTAP10-10, KRTAP10-11, KRTAP10-12, KRTAP11-1, KRTAP12-1, KRTAP12-2, KRTAP12-3, KRTAP12-4, KRTAP13-1, KRTAP13-2, KRTAP13-3, KRTAP13-4, KRTAP15-1, KRTAP16-1, KRTAP17-1, KRTAP19-1, KRTAP19-2, KRTAP19-3, KRTAP19-4, KRTAP19-5, KRTAP19-6, KRTAP19-7, KRTAP19-8, KRTAP20-1, KRTAP20-2, KRTAP20-3, KRTAP20-4, KRTAP21-1, KRTAP21-2, KRTAP21-3, KRTAP22-1, KRTAP22-2, KRTAP23-1, KRTAP24-1, KRTAP25-1, KRTAP26-1, KRTAP27-1, KRTAP29-1, KRTAP4-11, KRTAP4-12, KRTAP5-10, KRTAP5-11, KRT87P, KRTAP1-1, KRTAP1-3, KRTAP1-4, KRTAP1-5, KRTAP2-1, KRTAP2-2, KRTAP2-3, KRTAP2-4, KRTAP3-1, KRTAP3-2, KRTAP3-3, KRTAP4-1, KRTAP4-2, KRTAP4-3, KRTAP4-4, KRTAP4-5, KRTAP4-6, KRTAP4-7, KRTAP4-8, KRTAP4-9, KRTAP5-1, KRTAP5-2, KRTAP5-3, KRTAP5-4, KRTAP5-5, KRTAP5-6, KRTAP5-7, KRTAP5-8, KRTAP5-9, KRTAP6-1, KRTAP6-2, KRTAP6-3, KRTAP7-1, KRTAP8-1, KRTAP9-1, KRTAP9-2, KRTAP9-3, KRTAP9-4, KRTAP9-6, KRTAP9-7, KRTAP9-8, KRTAP9-9, KRT34, KRT35, KRT36, KRT37, KRT38, KRT81, KRT82, KRT83, KRT84, KRT85, KRT86, KRT222, KRT33A, KRT33B, KRTCAP2, KRTDAP, LALBA, PPL, PKP1, PKP2, PKP3, PKP4, JUP, PVALB, UPK1A, UPK1B, UPK2, UPK3A, UPK3B and UPK3BL; interestingly, interactions involving these behaved very much like negative interactions when we performed the ROC analysis below). To identify protein–protein relationships most supported by the TAP-MS observations, we derived a modified socioaffinity index[3,13], which is a sum of log-odds values that considers the frequency of protein pairs in the data set, either as bait–prey (spoke) or prey–prey (matrix) observations, and the overall frequency of proteins in the entire data set, which avoids the need to explicitly exclude 'sticky' proteins[13]. We modified the index to account for peptide coverage by first excluding those proteins where coverage was below 2% and then by using the coverage ratio (0–1) as counts in the socioaffinity calculation.

To benchmark these socioaffinity indices, we defined a set of positive interactions from protein interaction databases[55]: IntAct, BIND, BioGrid, DIP, Mint, HPRD and Uniprot. We required that interactions were independently reported at least three times either from different sources or by different methods indicating direct physical interactions. All of our selected databases register the interaction detection methods following the terms of OLS (http://www.ebi.ac.uk/ontology-lookup/[56]). Among all of the 'molecular interaction' terms, we selected 165 to be related to physical interactions. Our positive interaction set also excludes interactions detected by TAP-like methods. When more than 10 interactions in one publication share the same interaction ID or interactor, these were excluded from our set of positives. We defined a set of negative interactions from a set derived by analysis of high-throughput yeast two-hybrid studies[14]. Overall, we had 658,352 positive and 894,213 negative interactions.

We computed true-positive and false-positive rates (TPR and FPR) for decreasing socioaffinity thresholds. Inspection showed that we obtained very different curves depending on the nature of the protein pairs considered, with reverse tagging (that is, data when both proteins have been tagged) having different thresholds and generally better ROC plots (Supplementary Fig. 4), thus we considered the three classes (both-tagged, one-tagged or none-tagged) separately. Also because of a lack of complete reverse tagging, and general unreliable estimates

about the size of the interactome, we are unable to estimate the relative numbers of positives and negatives. For this reason, we took the stringent view of requiring both FPR (that is, fraction of negatives predicted by socioaffinity as positives) and false-discovery rate (FDR, or the fraction of predictions that are false positives) to be below a common threshold (0.1, 0.05, 0.01). Inspection showed that this gave a reasonable sensitivity but also avoided situations of a multitude of interactions involving common values that had FPR values near the threshold. We compared the coverage-weighted to binary (that is, protein present or not) counts for the socioaffinity calculation, and though the difference was marginal overall, inspection of the resulting networks showed a better resolution of sub-complexes known in the literature (for example, Exocyst, RPGR, see main text), which prompted us to use the weighted values.

*Clustering to identify putative complexes.* By considering each type of interaction (none, one or both proteins tagged) independently in the computation of the FPR and FDR values resulted in a non-monotonic relationship between FPR, FDR and socioaffinity indices. To take into account both FDR and socioaffinity during clustering we devised a score so that FDR is monotonically decreasing with respect to the new score:

$$\text{SA}_{\text{FDR}}(i,j) = \max_{(k,l):\text{FDR}(k,l) > \text{FDR}(i,j)} \text{SA}_{\text{FDR}}(k,l) + \text{SA}(i,j) - \min_{(k,l):\text{FDR}(k,l)=\text{FDR}(i,j)} \text{SA}(k,l)$$

The new score was computed iteratively starting with the interactions with the highest FDR value. For these interactions there are no protein pairs with higher FDR and the first term in the above equation is zero.

Protein complexes were predicted from the weighted PPI network using Hierarchical Clique Identification (HCI). The algorithm merges proteins in clusters based on their interaction scores using a hierarchical agglomerative clustering (HAC) approach that allows overlapping clusters. HCI starts by considering a weighted graph constructed from experimental data. In this graph, nodes correspond to clusters and edge weights are measures of similarity (for example, socioaffinity scores). Initially each protein is assigned to a cluster. At each iteration the algorithm selects the clusters to be combined and then updates the weights of the edges between clusters. For the selection of the clusters to be merged, an unweighted network is constructed by considering the edges with a weight equal to the maximum weight. Then the algorithm mines this network for maximal cliques, that is, cliques that are not contained in larger cliques. The extracted maximal cliques define the set of clusters to be combined. Nodes corresponding to merged clusters are removed from the network and new nodes are introduced for the new clusters. In general, linkage criteria similar to those used in HAC could also be used in HCI. In this study clusters are connected if the union of their members forms a clique and the weight of the edge connecting them is equal to the maximum weight of the edges connecting their non-common members. Formally, the linkage criterion between two clusters $X$ and $Y$ is given by:

$$d(X, Y) = \max_{x \in X \setminus Y, y \in Y \setminus X} d(x,y) \quad \text{if } \forall x \in X, \forall y \in Y (x,y) \in E$$

It is clear that only pairs of clusters that form a clique upon merging are connected in the new graph. This may result in the loss of the highest weighted edges of some nodes. This happens for instance when the best neighbour of a node is merged in a cluster, which is not fully connected with the node. To avoid merging of the node based on lower weighted edges, whenever a node loses its highest weighted edges it is removed from the graph. The entire procedure terminates when the graph becomes disconnected or when a score threshold is reached, which is the case here.

Except for the score threshold an additional filtering step was used to identify the sub-structure of the predicted complexes, if any. This was achieved by using Dirichlet process mixture (DPM) model[57]. DPM is a probability mixture model with an infinite number of components, which are mixed according to a stochastic process called Dirichlet process. DPM has been extensively used for data clustering due to its property that the stochastic process for mixture proportions almost surely produces a finite number of distributions. In DPM a cluster $i$ is described by a parametric distribution $f(\cdot \mid \theta_i)$, in our case Gaussian. The mixture model has the form $y_j \sim \sum_{i=1}^{K} \pi_i f(\cdot \mid \theta_i)$, where $K$ is the number of clusters and $\pi_i$ the mixture proportion of the distribution $f(\cdot \mid \theta_i)$. Moreover, the distribution parameters $\theta_i$ are drawn from a base distribution $G_0$. While the initial number of components $K$ is infinite, DPM ensures a finite number of components by properly selecting the mixture proportions $\pi_i$ from a dirichlet distribution.

For each cluster that was identified in the first step the distance matrix, consisting of the socioaffinity scores for all protein pairs in the cluster, was constructed. DPM was then used to identify sub-complexes from the distance matrix. Variational inference[58] was used to fit the DPM to the data and to identify the optimal clustering. If the DPM was consisting of a single component then the cluster was left intact. Otherwise, the branch of the dendrogram corresponding to the specific cluster was traced backwards to identify the subclusters that best match the DPM clustering.

**Gene enrichment analyses.** We extracted various gene/protein sets either from Gene Ontology or from Uniprot. From the latter we extracted complexes by identifying canonical gene names within Subunit/Complex descriptions for particular Uniprot accessions, and extracted genes related to genetic diseases from specific mutations linked to particular disease types (for example, BBS2) adding generic (for example, BBS) names where appropriate. To compute enrichment we used a Fisher exact test corrected for multiple testing where we estimated an

effective total of sets (in each class, for example, Complexes, Diseases, etc.) by considering sets sharing >80% overlapping genes/proteins to be the same set (that is, to avoid over-correction). A tool for computing this enrichment for human genes/proteins on these data sets (and others including the complexes determined previously) is available at http://getgo.russelllab.org.

**IFT-B sub-complex analysis.** *Sucrose density gradient centrifugation.* Sucrose gradients for density centrifugation were prepared in 2 ml centrifugation tubes. 250 μl of each concentration (20/17/14/11/8/5% sucrose) were discontinuously applied to the tube and overlaid with the pooled eluates from two individual SF-TAP purifications from a HEK293 cell line stably expressing IFT88-SF-TAP. After centrifugation at $166,000g_{\text{AV}}$ for 4 h in a swing-out rotor (Beckman TLS65), the gradient was fractionated by pipetting into 125 μl fractions[49]. The fractions were precipitated by methanol–chloroform and subjected to label-free quantification by mass spectrometry.

*EPASIS.* For protein complex destabilization the cleared lysates from HEK293 cells, stably expressing IFT88-SF-TAP, respectively, IFT27-SF-TAP were transferred to anti-FLAG M2 agarose (Sigma-Aldrich). After 1 h of incubation, the resin was washed three times using wash buffer (TBS containing 0.1% NP-40 and phosphatase inhibitor cocktails II and III, Sigma-Aldrich). For the SDS-destabilization of the protein complexes, the resin was then incubated 3 min with each concentration of SDS (0.00025, 0.0025, 0.005, 0.01 and 0.1%) in SDS-elution buffer (TBS containing phosphatase inhibitor cocktails II and III) at 4 °C. The flow through was collected and precipitated by methanol–chloroform. After every elution step a single wash step was performed. Subsequently to the SDS gradient, the remaining proteins were eluted from the resin by incubation for 3 min with FLAG peptide (200 μg ml$^{-1}$; Sigma-Aldrich) in wash buffer. The fractions were subjected to label-free quantification by mass spectrometry.

Statistical data analysis was carried out in R[59] by calculating the elution profile distance for each protein to the consensus profile for IFT-B1 and IFT-B2 (ref. 31). For each cell line, stably expressing IFT27, IFT88 or SF Control, six replicated EPASIS experiments were performed (108 measurements). Unique peptides with a minimum peptide length of seven amino acids were identified by searching against the forward and a reversed version of the database which indicates an average peptide false-positive identification rate of 0.17% for the experiments.

Without filtering proteins were detected for both, the forward and the reverse search, leading to an average indicated protein false-positive identification rate of 0.74 % (Supplementary Table 6). To reduce the number of false-positive protein identifications, proteins were considered as detected, if they were identified by at least two unique peptides, had a minimal MS/MS spectra count of three and were not flagged as contaminant by MaxQuant. Proteins that were detected in the control and the IFT27/IFT88 experiments, were tested using spearman's test and excluded from further considerations if they showed a significant ($P < = 0.05$) correlation between both experiments. Finally, proteins had to be present in at least 5/6 (83.33%) repeated experiments, resulting in a high confident list of 45 proteins for IFT27 and 19 proteins for IFT88 that were further analysed.

Protein intensities for all SDS concentrations of an experiment were combined and the values log2-transformed. To investigate the linear relationship between data points, regression lines determined by minimizing the sum of squares of the Euclidean distance of points to the fitted line ('orthogonal regression'). Correlations between repeated experiments were estimated using the Pearson correlation coefficient together with its 95% confidence interval. To investigate the safe isolation of elution profiles for different SDS concentrations, Spearman's correlation scores were calculated. Consensus profiles of known marker protein groups (Supplementary Table 6) were calculated by averaging the normalized cumulative intensities of the protein group per concentration step for all experiments. Elution profile distances (EPD) to consensus profiles were calculated for all detected proteins. A stepwise ($n = 1,000$) parameter search was performed to estimate the optimal EPD threshold to maximize the specificity and sensitivity to assign known sub-complex members to the consensus profile. To perform non-metric multidimensional scaling the elution profile distances were averaged across the replicated ($n = 6$) experiments and Euclidean distances between them calculated. A stable solution was estimated by using random starts and the best ordination (stress: 0.03 IFT27; 0.01 IFT88) selected.

**Rare variant discovery in the Syscilia cohort.** As part of our ongoing investigation of mutational burden in ciliopathies, we conducted bidirectional Sanger sequencing of coding regions and splice junctions of IFT-B encoding genes (*IFT172, IFT88, IFT81, IFT80, IFT74, IFT57, TRAF3IP1, IFT52, IFT46, IFT27, HSPB11, RABL5, IFT20* and *CLUAP1*) in a previously described ciliopathy cohort[60] according to standard methodology. The Duke University Institutional Review Board approved human subjects research, and DNA samples were ascertained following informed consent. PCR products were sequenced with BigDye Terminator v3.1 chemistry on an ABI 3,730 (Applied Biosystems), sequences were analysed with Sequencher (Gene Codes), and variants were confirmed by resequencing and visual assessment of chromatograms. Primer sequences and PCR conditions are available upon request.

**Identifying disruptive variants at complex interfaces.** We selected 10 IFT-B variants from the rare variant data set identified in ciliopathy cases (Syscilia cohort) as candidates to affect IFT architecture by an analysis of the structural features known or predicted for these subunits (Supplementary Data 7). We used the Mechismo[32] system and identified two variants at or near the interface (IFT27 p.R131Q and HSPB11 p.T41I) and a third on the surface of HSPB11 (p.R61S) but far away from this interface (and potentially a candidate to bind different proteins).

Though there were no other structures on which to confidently model IFT subunit interfaces, the fact that most of the subunits contain WD- or TPR repeats provided the means to use the location of common binding sites in these families to predict variants that might affect the interaction with a protein, even if the specific protein partner is not known. To do this we first defined domains by Pfam[61], TPRpred[62] and manual refinements were applied to define boundaries for WD-repeats in IFT172, and for TPR repeats in IFT88, and IFT172. We then aligned the sequences automatically coupled to manual editing and identified variants at or near the favourite binding site for WD-repeats (the top-side of the propeller[33]). For TPR repeats, binding site residues were defined by side-chain to peptide distances within a representative set of TPR repeats (from PDB codes 4n3a, 2lsv, 4buj, 1a17 and 1elr) superimposed and aligned using STAMP[63]. This alignment and a representative average structure showing the binding sites as depicted in Supplementary Fig. 7.

**Differential AP-MS to compare variants to wild-type proteins.** Protein complex comparison was done essentially as described before[11]. For SILAC labelling, HEK293T cells were grown in SILAC DMEM (PAA) supplemented with 3 mM L-glutamine (PAA), 10% dialyzed fetal bovine serum (PAA), 0.55 mM lysine and 0.4 mM arginine. Light SILAC medium was supplemented with $^{12}C_6$, $^{14}N_2$ lysine and $^{12}C_6$, $^{14}N_4$ arginine. Heavy SILAC medium was supplemented with either $^{13}C_6$ lysine and $^{13}C_6$, $^{15}N_4$ arginine or $^{13}C_6$, $^{15}N_2$ lysine and $^{13}C_6$, $^{15}N_4$ arginine. Proline (0.5 mM) was added to all SILAC media to prevent arginine to proline conversion[64]. All amino acids were purchased from Silantes. SF-TAP-tagged proteins and associated protein complexes were purified from HEK293T cells[10]. To this end, HEK293T cells, transiently expressing the SF-TAP-tagged constructs were lysed in lysis buffer containing 0.5% Nonidet-P40, protease inhibitor cocktail (Roche) and phosphatase inhibitor cocktails II and III (Sigma-Aldrich) in TBS (30 mM Tris-HCl (pH 7.4), 150 mM NaCl), for 20 min at 4 °C. After sedimentation of nuclei at 10,000$g$ for 10 min, the protein concentration of the cleared lysates was determined by Bradford before equal protein amounts were transferred to Strep-Tactin-Superflow beads (IBA) and incubated for 1 h. The resin was washed three times with wash buffer (TBS containing 0.1% NP-40, phosphatase inhibitor cocktail II and III). The protein complexes were eluted by incubation for 10 min in Strep-elution buffer (IBA). The eluted samples were combined before concentration using 10 kDa cut-off VivaSpin 500 centrifugal devices (Sartorius Stedim Biotech) and pre-fractionation using SDS–PAGE and in-gel tryptic cleavage[65].

**Yeast two-hybrid system.** A GAL4-based yeast two-hybrid system was used to screen for binary protein–protein interactions. Yeast two-hybrid constructs were generated according to the manufacturer's instructions using the Gateway cloning technology (Thermo Fisher Scientific) by LR recombination of GAL4-BD Gateway destination vectors with sequence verified Gateway entry vectors containing the cDNA's of selected bait proteins.

Constructs encoding full-length or fragments of bait proteins fused to a DNA-binding domain (GAL4-BD) were used as baits to screen human oligo-dT primed retinal, brain, kidney and testis cDNA libraries, a bovine random primed retinal cDNA library[66] or a library of human cDNA's from candidate and known ciliary proteins, fused to a GAL4 activation domain (GAL4-AD) or vice versa[67]. The yeast strain PJ69-4A, which carries the *HIS3* (histidine), *ADE2* (adenine), *MEL1* (α-galactosidase) and *LacZ* (β-galactosidase) reporter genes, was used as a host. Interactions were analysed by assessment of reporter gene activation based on growth on selective media (*HIS3* and *ADE2* reporter genes), α-galactosidase colorimetric plate assays (*MEL1* reporter gene), and β-galactosidase colorimetric filter lift assays (*LacZ* reporter gene).

**Ciliopathy genetic variants from UK10K data.** We downloaded ciliopathy patient data from the European genome-phenome archive (EGA), which consists of variants sequenced from 124 ciliopathy disease samples in 13 disease groups and 1 control. We mapped genomic variants using the Ensembl Variant Effect Predictor[68]. For each allele, we took the maximum allele frequency from those given by 1,000 Genomes[69] and Exome Aggregation Consortium (exac.broadinstitute.org) and in all instances only considered those lower than 1%. We computed the ratio between the frequency of variants in the UK10K and the 1,000 genome project, correcting these for the overall mutation rates in each set (to account for platform/variant calling differences). We calculated a $P$ value using a binomial test for each particular type of variation, compared (disease versus background) by Fischer's method. We considered genes having a ratio ≥ 2 and a $P$ value ≤ 0.01 as those significantly mutated in ciliopathies (either pooled or separately) relative to the healthy population (that is, in Figs 1 and 2). For Supplementary Data 5 we considered only homozygous/heterozygous (labelled) missense mutations with frequencies below 1%.

**Immunohistochemistry.** Unfixed kidneys and brains of 1-month-old Wistar rats were harvested and frozen in melting isopentane. Seven micrometre cryosections were cut and treated with 0.01% Tween in PBS for 20 min and subsequently blocked in blocking buffer (0.1% ovalbumin and 0.5% fish gelatin in PBS). After the blocking step, the cryosections were incubated overnight with the primary rabbit polyclonal antibody targeting GID8 (c20orf11 (N1C3), Genetex, cat. no. GTX106672; 1:100) or MKLN1 (Sigma-Aldrich, cat. no. HPA022817; 1:100) in combination with the monoclonal antibody GT335 (Adipogen, cat. no. AG-20B-0020-C100; 1:1,000), diluted in blocking buffer. Alexa Fluor 488- and 568-conjugated secondary antibodies were also diluted 1:500 in blocking buffer and incubated for 1 h in the dark. Staining of cell nuclei was performed with DAPI (1:8,000). Prolong Gold Anti-fade (Molecular Probes) was used for embedding the sections. Pictures were made with a Zeiss Axio Imager Z1 fluorescence microscope (Zeiss), equipped with a 63 × objective lens and an ApoTome slider. Images were processed using Axiovision 4.3 (Zeiss) and Adobe CS4 Photoshop (Adobe Systems). Procedures followed were in accordance with the ethical standards of the responsible committee on animal experimentation.

**Ciliary staining of HEK293T cells.** HEK293T cells were cultured in DMEM (PAA) supplemented with 10% fetal bovine serum and 0.5% penicillin/streptomycin. HEK293T cells were plated on glass slides coated with 0.01% poly-L-lysine (P8920 SIGMA) as described in the manufacturer's protocol. Slides were submerged in poly-L-lysine for half an hour and then rinsed twice with sterile MilliQ and allowed to dry for 1 h in the hood prior to use. Twenty-four hours after plating cells were starved for 48 h in 0.1% FCS (50% starvation medium and 50% 1 × PBS), 0.2% starvation medium, or full (10% FCS) medium. All conditions showed ciliated cells. IF images illustrate cells from the 0.1% starvation conditions. Cells were rinsed once with 1 × PBS at room temperature and then fixed in 2% PFA for 20 min and permeabilized with 1% Triton-X for 5 min. Cells were blocked in freshly prepared 2% BSA for 40 min and then incubated with the following antibodies for 1 h: a rabbit anti-ARL13B antibody (Proteintech, cat. no. 17711-1-AP; 1:500), a guinea pig polyclonal anti-RPGRIP1L antibody (SNC040, 1:300), and a monoclonal anti-acetylated tubulin antibody (clone 6-11-B1, Sigma-Aldrich, T6793; 1:1,000). Cells were stained with secondary antibodies for 45 min. The following secondary antibodies were used (all from Life Technologies/Thermo Fisher Scientific, Bleiswijk, The Netherlands; all diluted 1:500 in 2% BSA): anti-guinea pig IgG Alexa Fluor 647, anti-rabbit IgG Alexa Fluor 488, and anti-mouse IgG Alexa Fluor 568. DAPI stained the nucleus.

**TMEM41B ciliary phenotype.** *Cell line used.* Human kidney-2 (HK2) cells were cultured in DMEM F-12 5% FBS and supplied with ITS (SIGMA, I1884), 100 units per ml penicillin, and 100 μg ml⁻¹ streptomycin. Starvation in HK2 cells was achieved using DMEM F-12 without FBS for 24 h. Cells were grown at 37 °C with 5% $CO_2$.

*Immunofluorescence.* Cells were fixed in 4% paraformaldehyde. Blocking was performed in PBS-0.2% Triton X-100, 10% FBS. For cilia staining cells were starved for 24 h before fixation. Cilia were labelled with a rabbit anti-ARL13B antibody (Proteintech, cat. no. 17711-1-AP). Anti-FLAG was from Sigma (A8592). A total of 300 cells for mock, 150 cells for Clone A, 400 cells for Clone B were counted in the overexpression experiments. For siRNA interference experiments 70 cilia were measured for the negative control and 72 cilia were measured for TMEM41B depleted cells. Cilia length was measured using ImageJ (NIH). Cell confluence was comparable between overexpressing and control cells. The $P$ value was calculated with the $t$-Test **$P$ value < 0.01; ***$P$ value < 0.0001.

*Transfections.* TMEM41B was cloned in a p3XFLAG-CMVTM-14 expression vector (from Sigma-Aldrich E7908). HK2 cells were transfected using TransIT-LT1 Transfection Reagent (Mirus) according to the manufacturer's instructions and cells were collected 72 h after transfection both for WB and IF. As control, cells were treated with the Transfection reagent alone (Mock).

**RNAi.** ON-TARGET plus smart pool siRNAs against human TMEM41B and non-targeting control pool from Dharmacon were used at a concentration of 100 μM. The transfection reagent was INTERFERIN (409-10 from Polyplus). Silenced cells were used for IF analyses 96 h after transfections.

**3M syndrome proteins and relationship to cilia function.** *Cell culture.* Murine principle collecting duct (mpkCCD) clone 11 cells were grown with DMEM/Ham F12 1:1 vol/vol supplemented with 5 μg ml⁻¹ insulin; 50 nM dexamethasone; 60 nM sodium selenate; 5 μg ml⁻¹ transferrin; 1 nM triiodothyronine (T3); 2 mM glutamine; 10 μg ml⁻¹ epidermal growth factor (EGF); 2% fetal calf serum (FCS); 10% D-glucose; 20 mM HEPES, pH 7.4 and 10 μg ml⁻¹ ciproxin at 5% $CO_2$. Human fibroblasts were grown from skin biopsies in DMEM supplemented with 10% FCS and 1% P/S. Cells were incubated at 37 °C in 5% $CO_2$ to ∼90% confluence. Fibroblasts were serum starved for 48 h before fixation.

*Antibodies and reagents.* Antibodies used are mouse anti-CUL7 (clone Ab38, Sigma-Aldrich, C1743, diluted 1:500), mouse anti-acetylated tubulin (clone 6-11-B1, Sigma-Aldrich, T6793, diluted 1:20,000), rabbit anti-MKS1 (Proteintech 16206-1-AP, at 1:300) and rabbit anti-p38 MAPK Antibody (Cell Signaling, #9212, at 1:1,000).

Plasmid DNA transfection (1 microgram per well in a 6-well plate) was performed with Lipofectamine2000 (Thermo Fisher Scientific, 11668-019), according to the supplier's protocol. Opti-MEM (Thermo Fisher Scientific, 31985-062) was used to dilute the plasmids or mutant alleles. Human Wild-type plasmids were a kind gift from Dr Dan Hanson at University of Manchester, UK as previously published[70]: Myc-tagged CUL7, V5-tagged-OBSL1 and CCDC8.

Lipofectamine RNAimax (Thermo Fisher Scientific, 13778-075) was used for siRNA transfection of pooled siRNAs at a total final concentration of 20 nM, according to the supplier's protocol. Opti-MEM (Thermo Fisher Scientific, 31985-062) was used to dilute the ON-TARGETplus siRNA SMARTpools (Dharmacon): Non-targeting pool siCtrl (D-001810-10), mouse Ift88 (L-050417-00), mouse Obsl1 (L-058142-01), mouse Cul7 (L-054741-01), mouse Ccdc8 (L-067567-00).

*RT-qPCR.* RNA was isolated from cells using RNeasy Mini kit (Qiagen) an reverse transcription was performed using Siperscript III (Thermo Fisher Scientific). Quantitative real-time PCR was carried out using Sybr Green (Qiagen) and run in a MyiQ Single-color real-time PCR detection system (Bio-as Laboratories). Data were normalized to *Gapdh*. The mouse primer sequences (Sigma) used and concomitant annealing temperatures can be provided upon request. The ΔΔCT method was used for statistical analysis to determine gene expression levels.

*Western blot.* Protein lysates were prepared using RIPA lysis buffer. To correct for protein content BCA protein assay (Pierce) was performed. Anti-MAPK (1:1,000) was used as loading control in combination with Coomassie Blue staining. After SDS–PAGE separation and transfer, the PVDF membranes were blocked in 5% dried skim milk in TBS with 0.5% Tween. The primary antibody (or anti-CUL7 at 1:500) was incubated overnight at 4 °C. The secondary swine anti-rabbit and rabbit anti-mouse antibodies which are HRP conjugated (DAKO, dilution 1:2,000) were incubated for 1 h at RT. The ECL Chemiluminescent Peroxidase Substrate kit (Sigma, CPS1120-1KT) was used for development. Scans of the blots were made with the BioRad ChemiDoc XRS + device with Image Lab software 4.0.

*Immunofluorescence.* For immunostaining, mpkccd cells or fibroblasts were grown on glass coverslips and fixed for 5 min in ice-cold methanol and blocked 60 min in 1% BSA. Primary antibody incubations (mouse anti-acetylated tubulin 1:20,000, rabbit anti-MKS1, 1:300) were performed at 4 °C overnight in 1% BSA. Goat anti-mouse 488/rabbit 568 Alexa secondary antibody (Thermo Fisher Scientific, dilution 1:500) and DAPI incubations were performed for 2 h at RT. Coverslips were mounted in Fluormount G (Cell Lab, Beckman Coulter). Confocal imaging was performed using Zeiss LSM700 confocal laser microscope and images were processed with the ZEN 2012 software.

*Statistics.* P values were calculated of normally distributed data sets using a two-tailed Student's *t*-test, or one-way ANOVA with Dunnett's post hoc test, or two-way ANOVA with Bonferroni *post hoc* tests. Statistical analyses represent the mean of at least three independent experiments; error bars represent s.e.m. or indicated otherwise.

**Data availability.** Interaction, and complex data are available and http://landscape.syscilia.org/. Additionally, the protein interactions from this publication have been submitted to the IMEx (http://www.imexconsortium.org) consortium through IntAct (http://www.ebi.ac.uk/intact/) and assigned the identifier IM-25054.

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

## Acknowledgements

We thank the patient and parents for participation in research. We thank Gisela Slaats for technical assistance, the Syscilia consortium members for helpful scientific discussions, Colin Johnson for access to the siRNA datasets, and the Cell Microscopy Center Utrecht for Imaging assistance. The research leading to these results has received funding from the European Community's Seventh Framework Programme FP7/2009 under grant agreement no: 241955, SYSCILIA (to G.A., P.L.B, O.E.B., T.J.G, M.A.H., N.K., H.K., H.O., U.W., F.K., B.F., R.H.G., M.U., R.B.R. and R.R.), FP7 grant agreement no. 278568, PRIMES (to M.U and K.B.); the Dutch Kidney Foundation 'Kouncil' (CP11.18 to H.H.A., P.L.B., R.H.G. and R.R.); the Netherlands Organisation for Scientific Research (Veni-016.136.091 to E.v.W., Veni-91613008 to H.H.A., and Vici-865.12.005 to R.R.); the Foundation Fighting Blindness (grant C-CMM-0811-0546-RAD02 to R.R., and grant C-CMM-0811-0547-RAD03 to H.K. and E.v.W.); NIH grants DK075972 and HD042601 (N.K.); DK072301 (N.K. and E.E.D); and EY021872 (E.E.D). H.K. and E.v.W. acknowledge 'Stichting Nederlands Oogheelkundig Onderzoek', 'Stichting Blindenhulp', 'Stichting Researchfonds Nijmegen', 'Landelijke Stichting voor Blinden en Slechtzienden', and the Netherlands Organisation for Health Research and Development (ZonMW E-rare grant 40-42900-98-1006). M.B., Q.L. and R.B.R. are supported by the Excellence Initiative Cell Networks, Germany Science Ministry. N.K. is a distinguished Jean and George Brumley Professor. B.F. acknowledges support from the Telethon Foundation (TGM11CB3). M.U. was supported by the Tistou & Charlotte Kerstan Stiftung.

## Author contributions

R.R. and M.U. conceived the overall project. K.B., J.v.R, Q.L., K.K., M.U., K.U., P.A.T., C.G., R.H.G., R.B.R. and R.R. led the data generation and processing. K.B., Q.L., K.K., H.H.A., S.E.C.v.B., M.J.B., T.B., E.B., K.L.M.C., E.E.D., G.D., K.H., L.H., N.H., D.I., D.J., I.J.L., B.L., S.J.F.L., D.A.M., C.L.M., D.M., M.M., T.M.N., M.M.O., M.R., S.R., P.A.T., Y.T., G.T., T.J.v.D., E.V., J.W., Y.W. and K.M.W. performed experiments, M.U., R.B.R., R.R., G.A., P.L.B., O.E.B., T.J.G, M.A.H., N.K., H.K., H.O., UK10K, E.v.W., U.W., F.K., B.F. and R.H.G. Analysed and interpreted data. K.B., J.v.R., Q.L., K.K., R.H.G., R.B.R. and R.R. wrote the paper with input from all authors.

## Additional information

**Competing financial interests:** The authors declare no competing financial interests.

## UK10K Rare Diseases Group

Saeed Al-Turki[21,22], Carl Anderson[22], Dinu Antony[13], Inês Barroso[21], Jamie Bentham[23], Shoumo Bhattacharya[23], Keren Carss[21], Krishna Chatterjee[24], Sebahattin Cirak[25], Catherine Cosgrove[23], Petr Danecek[21], Richard Durbin[21], David Fitzpatrick[26], Jamie Floyd[21], A. Reghan Foley[25], Chris Franklin[21], Marta Futema[27], Steve E. Humphries[27], Matt Hurles[21], Chris Joyce[21], Shane McCarthy[21], Hannah M. Mitchison[13], Dawn Muddyman[21], Francesco Muntoni[25], Stephen O'Rahilly[24], Alexandros Onoufriadis[13], Felicity Payne[21], Vincent Plagnol[28], Lucy Raymond[29], David B. Savage[24], Peter Scambler[13], Miriam Schmidts[13], Nadia Schoenmakers[24], Robert Semple[24], Eva Serra[21], Jim Stalker[21], Margriet van Kogelenberg[21], Parthiban Vijayarangakannan[21], Klaudia Walter[21], Ros Whittall[27], Kathy Williamson[26]

[21]The Wellcome Trust Sanger Institute, Wellcome Trust Genome Campus, Hinxton CB10 1HH, Cambridge, UK. [22]Department of Pathology, King Abdulaziz Medical City, Riyadh, Saudi Arabia. [23]Department of Cardiovascular Medicine and Wellcome Trust Centre for Human Genetics, Roosevelt Drive, Oxford, OX3 7BN, UK. [24]University of Cambridge Metabolic Research Laboratories, and NIHR Cambridge Biomedical Research Centre, Institute of Metabolic Science, Addenbrooke's Hospital, Cambridge, CB2 0QQ, UK. [25]Dubowitz Neuromuscular Centre, UCL Institute of child health & Great Ormond street Hospital, London, WC1N 3JH, UK. [26]MRC Human Genetics Unit, MRC Institute of Genetic and Molecular Medicine, at the University of Edinburgh, Western General Hospital, Edinburgh, EH4 2XU, UK. [27]Cardiovascular Genetics, BHF Laboratories, Rayne Building, Institute Cardiovascular Sciences, University College London, London WC1E 6JJ, UK. [28]University College London (UCL) Genetics Institute (UGI) Gower Street, London, WC1E 6BT, UK. [29]Department of Medical Genetics, Cambridge Institute for Medical Research, University of Cambridge, CB2 2XY, UK.

