## [Peer review file · Nature Communications]

Transferred manuscripts:

Reviewers' Comments:

Reviewer #2 (Remarks to the Author)

The authors have not been able yet to address the abundance and stoichiometry of the bait and prey proteins in the current study. This may complement the socioaffinity and clique determinations and further buttress the hypothesis that these methods do indeed indicate complexes that represent those inside the cell. Would the authors consider doing this for one or more of their complexes?

The authors have also not yet been able to address whether the ciliary proteins they characterize are localized to cilia in organs. The authors of course are experts in cilia and are well aware of the motile cilia of the ventricles of the brain, oviduct, and trachea, all with multi-cilia. These can be labeled using standard methods for immunolabeling of cells in organs. Again should the authors consider for their gold standard proteins and those featured in the Figures in the manuscript at least a few they can show are in fact localized to cilia in situ?

The supplementary tables gather most of the primary data and even with the new tables; it may not be obvious how to navigate through these when limited by the information in the Table legends. Again if one starts with the supplementary tables it may not be obvious how the complexes featured in the paper were deduced. A selection based on landscape potential and selected sub complexes may not be obvious from the supplementary tables.

One possibility for the authors' consideration is to gather those proteins that have been discovered here that localize to the first time to cilia in organs, then gather the complexes and sub complexes that they deduce to associate with these proteins. These could be the ones also supplemented for the requested abundance and stoichiometry.

Reviewer #3 (Remarks to the Author)

All criticisms have been adequately addressed by the authors.